# Tetanus Vaccine Knowledge, Beliefs, and Attitudes Among Syrian Pregnant Women in Türkiye: A Qualitative Study

**DOI:** 10.3390/healthcare13030302

**Published:** 2025-02-02

**Authors:** Zeynep Meva Altaş, Bayan Abdulhaq, Mehmet Akif Sezerol, Selma Karabey

**Affiliations:** 1Maltepe District Health Directorate, Istanbul 34841, Türkiye; 2Department of Public Health, International School of Medicine, Istanbul Medipol University, Istanbul 34815, Türkiye; 3School for International Training, Brattleboro, VT 05302, USA; bayan.abdulhaq@sit.edu; 4Epidemiology Program, Institute of Health Sciences, Istanbul Medipol University, Istanbul 34815, Türkiye; masezerol@gmail.com; 5Department of Public Health, School of Medicine, Istanbul Medipol University, Istanbul 34815, Türkiye; 6Sultanbeyli District Health Directorate, Istanbul 34935, Türkiye; 7Department of Public Health, Istanbul Faculty of Medicine, Istanbul University, Istanbul 34093, Türkiye; skarabey@istanbul.edu.tr

**Keywords:** migrants, tetanus vaccination, pregnant women

## Abstract

**Background:** In this qualitative study, we aimed to evaluate the knowledge, beliefs, and attitudes of Syrian pregnant women living in Türkiye toward tetanus vaccination, vaccine hesitations, and the perceived benefits and harms of the tetanus vaccine. **Methods:** In face-to-face, in-depth interviews, an open-ended, semi-structured interview guide was used by an interviewer and translator who spoke Arabic, which is the mother tongue of the participants. Codes, themes, and subthemes were created. **Results:** In the study, face-to-face in-depth interviews were conducted with a total of 16 pregnant women. The median age of the pregnant women was 26.5 years (18.0–41.0). The median time since they arrived in Türkiye as migrants was 8.5 years (3.0–10.0). The themes were “Beliefs about vaccines”, “Information about vaccines”, “Knowledge, beliefs and attitudes about tetanus vaccine”, “Reasons for vaccination desire”, “Reasons for not being vaccinated”, and “Problems experienced while receiving health services”. Pregnant women mostly mentioned that vaccines have benefits. Most of the women stated that tetanus vaccine is needed to protect from diseases. There were no women thinking that vaccines are harmful. Some of them only mentioned the minor side effects observed after vaccination. Participants mentioned that they get information about vaccines from relatives such as family and friends, health professionals, television, and social media. However, some women talked about the fact that they had no knowledge about vaccines. A lack of knowledge and not visiting the health institution were the reasons for not being vaccinated. **Conclusions:** Although participants mostly believed that vaccination is necessary and has benefits, some women had no idea about the exact purpose of vaccines. This lack of knowledge may cause vaccine hesitancy and refusal.

## 1. Introduction

Vaccination is one of the most successful public health interventions. According to the World Health Organization (WHO), immunization services prevent 3.5–5 million deaths each year from diseases such as diphtheria, tetanus, whooping cough, influenza, and measles. Vaccination helps people of all ages live longer and healthier [1].

Vaccine refusal and hesitancy are among the greatest obstacles to the success of vaccination services. Anti-vaccination sentiments are on the rise worldwide. According to the WHO, vaccine hesitancy was one of the 10 global health threats of 2019 [2]. As immunization programs continue to expand, understanding and addressing vaccine hesitancy is becoming important for the successful implementation of immunization services. A lack of information or misinformation, concerns about vaccine safety (side effects, etc.), distrust of vaccine policies, the influence of anti-vaccine posts on social media, religious and personal beliefs, and distrust of vaccine manufacturers are some of the reasons for individuals’ vaccine hesitancy [3,4].

Vaccine hesitancy poses significant risks not only to individuals with vaccine hesitancy but also to society. Delaying and refusing to vaccinate prevents the community from reaching herd immunity levels, thus allowing a vaccine-preventable organism to circulate in that community. As a consequence, the likelihood of outbreaks due to the infectious agent increases [5].

On the other hand, situations such as conflicts and forced migration that destroy public health infrastructure and disrupt preventive measures such as vaccination should also be mentioned. Migration is the movement of people from where they live to a different place for economic, social, political, and cultural reasons. According to the latest data of the Turkish Directorate of Migration Management dated 16 November 2023, 3,246,178 registered Syrian individuals are living in Türkiye [6]. Although the Turkish Ministry of Health has developed various health services, especially for Syrian migrants, migrants may still encounter some challenges in accessing health services. The reasons mostly arise from language barriers, low educational levels, and economic problems. Pregnant women are one of the most important populations to whom health services should be successfully delivered. The disruption of basic health services such as vaccination in pregnant women is important for the health of both the mother and the baby. Therefore, it is extremely important to evaluate vaccination services for pregnant migrant women.

The vaccination of pregnant women is one of the main strategies recommended by the WHO to eliminate tetanus disease [7]. It is necessary to prevent both maternal and neonatal tetanus disease [8]. Receiving at least two doses of the tetanus vaccine is estimated to reduce mortality from neonatal tetanus by 94% according to a previous study [9]. However, in the literature, the tetanus vaccine uptake levels of pregnant migrant women living in Türkiye were reported to be low [10,11]. The tetanus vaccination is one of the health services offered free of charge to pregnant migrant women in Türkiye. According to the Strategic Advisory Group of Experts (SAGE), preventing vaccine hesitancy begins with a deeper understanding of its determinants and the associated challenges [12]. In this context, the present study aimed to explore the knowledge, beliefs, and attitudes of pregnant migrant women living in Türkiye regarding tetanus vaccination, with a focus on their vaccine hesitations and perceptions of the benefits and potential harms associated with the tetanus vaccine.

## 2. Materials and Methods

### 2.1. Study Type, Design and Participants

For this qualitative type of study, in-depth interviews were used as the data collection method. Data collection was terminated after it was determined by the researchers that data saturation had been reached. Thus, in-depth interviews were conducted with 16 participants. The participants were selected among pregnant women registered in the Migrant Health Center (MHC), which is a primary health care institution serving under the District Health Directorate in Istanbul. Before the interviews, the participants were informed about the study (aim, topic, researchers), and their consent was obtained. For travel and other expenses, each participant received a TRY 20 grocery voucher prior to the interview. The inclusion criteria for this study were Syrian migrant pregnant women aged 18 years or older who visited the MHC for pregnancy follow-ups or other healthcare services. Participants were selected among those who were informed about the study and provided their consent to participate.

In face-to-face interviews, an open-ended, semi-structured interview guide was used by an interviewer and translator who spoke Arabic, which is the mother tongue of the participants. After collecting sociodemographic and maternal information, such as age, education level, history of low-risk pregnancies, and clinical features related to pregnancy, participants were asked about their opinions on tetanus and other vaccines, their sources of information, perceived benefits and harms of vaccines, and barriers to vaccination. The interviews lasted between 20 and 30 min. The interviewer was a 36-year-old female nurse who has been working as a coordinator nurse at the MHC for 1.5 years. The nurse had experience in qualitative research. Before the interviews, she was briefed about the study objectives by the authors. The interviewee nurse provides coordination services at the MHC for pregnancy follow-ups/maternity follow-ups, infant/child follow-ups, and vaccinations. The interviews were conducted in a pregnancy follow-up room located in the MHC, which was quiet and clean, with no outside interference. The study was conducted in accordance with the COREQ guidelines.

### 2.2. Data Analysis

In the evaluation of the data, the stages of access to healthcare by Levesque et al. were followed [13,14]. A thematic analysis was conducted using a framework approach comprising seven steps to systematically analyze qualitative data [13]:Transcription: All interviews were audio-recorded and later transcribed. During the in-depth interviews, the interviewer used the Turkish version of the interview guide. A translator facilitated communication between the interviewer and the participants throughout the sessions. Consequently, the audio recordings included the translator’s Turkish interpretation of both the questions and participants’ responses. The researchers manually transcribed the recordings into Turkish and then independently translated the transcripts into English. To ensure accuracy, both researchers cross-checked the translations.Familiarization: The English translations of the interviews were reviewed to identify primary topics and potential codes and themes for analysis.Coding, Categorization, and Theme Development: While creating the code list, most of the items in Levesque et al.’s model were accepted as deductive codes or categories. Additionally, to cover situations not included in Levesque et al.’s model, new codes were created inductively by reading the content.Analytic Framework Construction: Levesque et al.’s model stages were followed, assessing complementary individual and institutional dimensions at each stage.Applying the Analytical Framework: Interviews were coded by two researchers. The coding process was conducted in accordance with COREQ guidelines. During the first stage (open coding), a total of 42 codes were generated. In the second stage (merging codes and transforming them into themes), minor disagreements arose between the two coders regarding 7 codes. However, these discrepancies were resolved through mutual discussion and consensus between the coders. As a result, no third stage or involvement of an external adjudicator was necessary.Data Charting: Atlas.ti was used to generate a report detailing the content and codes for each interview to facilitate analysis.Data Interpretation: Thematic analysis was conducted using the reports, with themes and subthemes independently identified by two researchers (ZMA, MAS).

In the results section, some quotes from the participants’ answers were included.

### 2.3. Ethics

Ethics committee approval for the study was obtained from the Istanbul Medipol University Non-Interventional Clinical Research Ethics Committee under decision number 580 on 13 July 2023.

## 3. Results

In the study, face-to-face in-depth interviews were conducted with a total of 16 pregnant women. The median age of the pregnant women was 26.5 years (18.0–41.0). The median time since they arrived in Türkiye as refugees was 8.5 years (3.0–10.0). Three pregnant women were currently pregnant for the first time and had no other children. The median gestational week was 28.5 weeks (22.0–40.0). Four pregnant women had a history of miscarriage, while two pregnant women had a history of stillbirth. All pregnant women, except one, were having their current pregnancy control visits regularly. Sociodemographic characteristics of the 16 pregnant women who were interviewed are shown in Table 1.

In the thematic analysis of the texts translated into Turkish after transcription, six different themes and sixteen subthemes were produced. The themes are “Beliefs about vaccines”, “Information about vaccines”, “Knowledge, beliefs and attitudes about tetanus vaccine”, “Reasons for vaccination desire”, “Reasons for not being vaccinated”, “Barriers to accessing health services”. The themes and subthemes are shown in Table 2.

### 3.1. Theme 1: Beliefs About Vaccines

#### Subtheme 1: Perceived Benefits About Vaccines

Pregnant women predominantly emphasized that vaccines are essential and provide numerous benefits. They expressed that vaccines play a crucial role in protecting both their own health and the health of their babies. Many participants highlighted that receiving vaccines during pregnancy helps prevent various diseases, ensuring a safer pregnancy and contributing to the well-being of their newborns. This indicates a general understanding of the preventive nature of vaccinations and their significance in maternal and child health. The examples of the statements of pregnant women are written below:


*“It protects them from diseases, protects them in their later years, strengthens their immune system and is a safe thing for children.”*
(P2)


*“Vaccinations are very good and important, especially for pregnant women, some women may not have a good diet, but they benefit from vaccines. It is also useful for children, it is useful for fewer illnesses or when there is a vaccination, I want to go immediately.”*
(P9)


*“When we get these vaccines to prevent measles or some other diseases, we don’t get sick…. İt strengthens the immune system.”*
(P13)

None of the women in the study expressed the belief that vaccines are harmful. However, some participants mentioned experiencing minor side effects following vaccination. These side effects were described as mild and temporary, such as soreness at the injection site or slight discomfort. Despite these experiences, the women did not perceive these side effects as significant enough to outweigh the benefits of vaccination. This finding reflects a generally positive attitude toward vaccines and an understanding of their role in preventing serious health risks during pregnancy.


*“There will be no effect because all my children have received their vaccinations and they are all healthy now.”*
(P1)


*“Fever can rise after vaccination, but it’s normal. It usually goes away in a few hours.”*
(P13)

### 3.2. Theme 2: Information About Vaccines

Most women demonstrated a basic understanding of vaccines, recognizing their general purpose and benefits. However, some women admitted having no knowledge about vaccines, indicating a gap in awareness among certain individuals. When asked about their sources of information, pregnant women mentioned close relatives, such as family members and friends, as well as healthcare professionals, who were often regarded as trustworthy and authoritative sources. Additionally, they mentioned accessing information through mass media, including television, and digital platforms like social media. This highlights the diverse ways in which women gather information about vaccines, while also emphasizing the potential for misinformation, particularly from less regulated sources such as social media and television.


*“I don’t think anything because I have not received any vaccine so far….. I have no information about vaccines…. No, I have never heard of it”*
(P4)


*“My mother tells me, my relatives tell me, I learned from my family…. Doctors recommended vaccines”*
(P16)

#### Subtheme 1: Need for İnformation

Some pregnant women mentioned that they have the need to be informed about vaccines. They stated that they lack information about the post-vaccination process (such as side effects and how long is the effect of protection). Some quotes are given below:


*“It’s definitely helped me. But there’s definitely more to it than that. I don’t know because I haven’t read them.”*
(P12)


*“Why was this vaccine given, how long will it protect, what are its benefits? …. I would like to know these.”*
(P6)

### 3.3. Theme 3: Knowledge, Beliefs, and Attitudes About Tetanus Vaccine

The majority of the participants had no knowledge about tetanus disease. Additionally, the majority of women indicated that they had received the vaccines recommended to them during their pregnancies. This demonstrates a level of trust in healthcare professionals and their recommendations, despite the lack of detailed knowledge about tetanus among many participants.

#### 3.3.1. Subtheme 1: Knowledge About Tetanus Disease

Most pregnant women reported that they had not been informed about tetanus disease, reflecting a significant gap in awareness about this health condition. Among those who were familiar with tetanus, many described it as a serious disease that has risks if left untreated.


*“It’s a dangerous disease. But I don’t know much about it.”*
(P10)


*“It is a dangerous disease for pregnant women and babies. Vaccinations are taken in such cases and should be followed during pregnancy.... I don’t know how it is transmitted but”*
(P5)


*“We need to keep them away from germs, and when children fall or get injured, we need to clean the wound. Or we need to give tetanus vaccine.”*
(P8)

#### 3.3.2. Subtheme 2: Knowledge and Beliefs About Tetanus Vaccine

The majority of respondents highlighted the crucial role of the tetanus vaccine, underscoring its importance in safeguarding both maternal and neonatal health from life-threatening complications. One woman specifically highlighted that the tetanus vaccine is necessary to prevent physical disabilities in children.


*“Some people get tetanus vaccine to protect them from diseases, some people get tetanus vaccine when they are injured….. The most important thing is to get the vaccine to prevent tetanus disease”*
(P14)


*“We need to be vaccinated. To prevent physical disabilities in children. Doctors talked about them.”*
(P6)

#### 3.3.3. Subtheme 3: Attitudes Towards Tetanus Vaccine

This subtheme focuses on the participants’ attitudes toward the tetanus vaccine, highlighting their perceptions and acceptance levels. Attitudes were shaped by several factors, including personal beliefs about the vaccine’s benefit and trust in the advice of healthcare workers.


*“I received pregnancy vaccinations, I learned from the doctors and nurses and and I also ensure that my children are regularly vaccinated.”*
(P3)


*“I’m taking the tetanus vaccine now that I’m pregnant, and I’m doing it for the baby’s health.”*
(P14)


*“Vaccinations are useful and good, I received all of them, I received all my vaccinations.”*
(P15)

### 3.4. Theme 4: Reasons for Vaccination Desire

The majority of women mentioned that they received the tetanus vaccine primarily for the protection of their own health. Many also emphasized that they chose to get vaccinated for the well-being of their baby, recognizing the vaccine’s positive role in their child’s health. Most participants stated that the recommendations given by healthcare professionals played a crucial role in their decision to receive the vaccines. This highlights the essential role of healthcare professionals in promoting vaccination during pregnancy.

#### 3.4.1. Subtheme 1: Health for Pregnant Women

As mentioned in the previous theme, most pregnant women, not fully understanding what tetanus disease entails, expressed that the tetanus vaccine is necessary for maternal health, stating that it protects the mother from diseases. However, they were unable to specify the exact nature of the disease, the symptoms, or which illnesses it protects against, and provided a general response instead.


*“I felt that my energy has increased or that the vaccine will protect me from diseases, I have such a belief now.”*
(P3)


*“It strengthens you to avoid diseases.”*
(P16)

#### 3.4.2. Subtheme 2: Health for Baby/Children

Similarly, for their children, many pregnant women acknowledged the importance of vaccination but were unable to specify the diseases or symptoms the vaccines protect against. They generally understood that vaccines are necessary to keep their children healthy but lacked detailed knowledge about the specific benefits.


*“We vaccinate children to help them develop or to prevent them from infecting anyone if they get sick.”*
(P9)


*“It protects children from diseases. In the past, children used to have physical disabilities, but not anymore due to vaccinations.”*
(P8)

#### 3.4.3. Subtheme 3: Recommended by Health Workers

This subtheme highlights the role of healthcare workers in guiding pregnant women on the tetanus vaccination. Many participants expressed trust in the recommendations provided by healthcare workers, believing that healthcare workers prioritize the well-being of their patients. The assurance that the vaccine would not be suggested unless it had clear benefits shows a significant source of confidence for the participants in their decision process.


*“The doctor informed me that the vaccine should be administered at the end of the fifth month and the beginning of the sixth month…. Healthcare professionals generally do not recommend anything that could harm the body; if there were no benefits, they would not suggest it.”*
(P8)

### 3.5. Theme 5: Reasons for Not Being Vaccinated

Some women talked about that they had no knowledge about vaccines. Lack of knowledge was the reason for lack of vaccination for them. Some pregnant women said that when they first came to the country as refugees, they could not apply to a health institution because they did not have an identity card, and that this was the reason why they were not examined and vaccinated.


*“I have never received a vaccine so far… I would still like to have information. If someone tells me about getting vaccinated, if the information is useful, maybe I might change my mind.”*
(P4)


*“I was not vaccinated in the first pregnancy… I could not get an ID card in Türkiye, so I was not vaccinated in the previous pregnancy… I could not be admitted to the hospital.”*
(P5)

### 3.6. Theme 6: Barriers to Accessing Health Services

While the majority of participants mentioned that they did not face any issues accessing healthcare services, some highlighted barriers such as language difficulties, the lack of recommendations from healthcare workers, insufficient social support, and transportation challenges. Pregnant women frequently stated that they did not have any language problems while receiving health services. Some pregnant women mentioned that they communicated with health workers with the help of the hospital’s translator or with the help of their partner while receiving services in health institutions.


*“I don’t speak any Turkish. I have a relative who translates for us… And during previous labor, a translator helped over the phone. It was difficult for me.”*
(P1)


*“No, I don’t have any problems, but I have difficulties because of the language, because I don’t speak Turkish.”*
(P15)

Some pregnant women mentioned that during their previous pregnancies, they were not adequately informed by healthcare professionals about the importance of vaccines, including the tetanus vaccine. This issue was a barrier to accessing healthcare services and contributed to a gap in their knowledge about the benefits and potential risks associated with vaccination.


*“The private doctors did not recommend vaccination during pregnancy (She explains about previous pregnancy).”*
(P3)

Although almost all of the pregnant women stated that they did not experience problems in accessing health services in terms of transportation and social supporters accompanying them, one pregnant woman mentioned that she experienced problems in terms of transportation and social support. Another woman said that when she wanted to go to her check-ups when she was pregnant, she had to walk because there was no public transportation, so she had difficulties. However, this did not hinder her check-ups.


*“I have a small child, have no one to accompany me and the house is far away, and I have to come on foot.”*
(P2)


*“There is no car, there is a long distance from my house. there is no public transportation, the house is far away, I come on foot, it’s a little difficult for me… Yet, I came all my controls.”*
(P14)

## 4. Discussion

Pregnancy is a period that requires special care, and tetanus vaccination is one of the key components of antenatal care [15]. Migrants often face challenges in accessing healthcare services, including vaccination [14,15,16,17,18]. A recent systematic review revealed that migrants in the United States have lower vaccination rates compared to the native population [19]. Understanding knowledge about the tetanus vaccine, as well as the reasons for being vaccinated or not, is crucial for developing effective public health strategies. In this context, this qualitative study explored the beliefs, knowledge, and attitudes of Syrian migrant women living in Türkiye regarding the tetanus vaccine.

In this study, none of the women stated that vaccines are harmful. The majority of participants stated that vaccines are essential for maternal health and the health of the child. They frequently emphasized that vaccines provide protection against diseases. Similarly, a qualitative study conducted with pregnant women in the United Kingdom found that most women believed receiving the recommended vaccinations during pregnancy was important and expressed positive attitudes toward vaccines. They often mentioned that vaccines are necessary to protect their babies and/or themselves, which aligns with the findings of our study [20].

In this study, pregnant women mentioned receiving information about vaccines from healthcare professionals, relatives, television, and social media. However, the accuracy of the information presented on social media and television should be carefully monitored. The majority of women in our study had basic knowledge about vaccines. However, it is concerning that some women had no information about vaccines. Additionally, a lack of knowledge was frequently cited as a reason for not getting vaccinated. Similarly, a study in the literature on influenza vaccination among pregnant women reported that knowledge about the vaccine was associated with vaccination attitudes [21]. In another study, the level of knowledge about vaccination was found to be lower among unvaccinated pregnant women compared to vaccinated pregnant women [22]. A lack of knowledge about vaccinations in pregnant women has been reported even among health professionals [23]. To prevent a lack of information, which is one of the barriers to vaccine hesitation [24], healthcare professionals should provide information about the intervals at which vaccines can be administered during pregnancy periods, possible side effects after vaccination, and protection rate and duration with up-to-date data. To prevent vaccine hesitancy, the communication skills of healthcare professionals can be improved to provide counseling about vaccination to pregnant women. This can help to solve the problem that women mentioned in our study that they need to be informed about vaccines.

The majority of pregnant women in our study cited protecting the health of mothers and babies, as well as receiving a vaccine recommended by healthcare professionals, as the main reasons for tetanus vaccination. In one study, it was reported that a direct recommendation significantly influenced immunization, leading to a 107-fold greater likelihood of vaccination [25]. However, another study found that 22.4% of women reported that physicians did not directly offer the vaccine to them [26]. Both our study and the literature suggest that vaccine recommendations from healthcare professionals positively impact vaccination rates among pregnant women, likely due to trust in healthcare providers. According to a recent systematic review, strong recommendations from physicians were identified as a trusted source of information for migrants [19]. For this reason, it is essential for healthcare professionals to inform pregnant women about the vaccinations offered as part of preventive health services and to provide appropriate counseling. Additionally, interventions such as reminder messages, phone calls, and engaging and persuasive brochures or posters can be planned, especially for disadvantaged and high-risk groups, such as migrants.

The social or transport barriers encountered by pregnant migrant women may cause problems in receiving health services. According to the literature, physical or logistical barriers have been reported to prevent the use of vaccine services [19]. In our study, these barriers were mentioned by the women as language difficulties, transportation problems, and the need for others (relatives, partners, etc.) to accompany them in health institutions. Language-related difficulties are frequently reported in the literature among the problems experienced by migrants in accessing health services [27,28]. MHCs have been established in places where Syrians live densely to provide preventive health services and basic health services to Syrians in Türkiye more effectively and efficiently, to overcome problems arising from language and cultural barriers, and to increase access to health services. It underscores the importance of culturally sensitive health communication and the need for healthcare personnel who can communicate in multiple languages. The awareness and accessibility of MHCs can be increased to improve immunization services for pregnant migrant women.

The limitation of the study is that the study was conducted with refugee pregnant women from a single MHC. In addition, social desirability bias can exist in the study. Also, the fact that the interviews were conducted in Arabic and then translated into Turkish and then into English for the printing process may have caused a minimal loss in the reflection of the women’s original statements in the findings.

Our study allowed an in-depth and multifaceted evaluation of the reasons for not vaccinating pregnant women who are not vaccinated against tetanus. Since it reveals the beliefs and attitudes of refugee pregnant women about vaccines, it would make important contributions to the literature in this field.

## 5. Conclusions

Our main findings indicate that Syrian pregnant women in Türkiye mostly lack broad knowledge about tetanus disease and tetanus vaccination. Although they mostly believe that vaccination is necessary and has benefits, some women had no idea about the exact purpose of vaccines. This lack of knowledge may cause vaccine hesitancy and refusal; it may even cause individuals to hold back from accessing other basic health services.

The study demonstrated the importance of providing information about the benefits of tetanus vaccines to pregnant women by healthcare professionals within the scope of preventive health services and providing counseling services appropriately. The study showed that healthcare professionals are trusted by migrant women as they are seen as credible sources of information. Interventions such as reminder messages, phone calls, attractive and persuasive brochures, and posters can be planned especially for disadvantaged and high-risk groups such as refugees.

The priority in health interventions and policies to prevent and reduce tetanus vaccine refusal in refugee pregnant women should be understanding the source of vaccine refusal and resolving potential barriers to vaccine refusal. In this context, the role of qualitative studies may be crucial. Further qualitative studies involving multiple centers may provide more comprehensive data in this regard. Interventions aiming to increase knowledge about immunization and the tetanus vaccine among refugee pregnant women are needed.

## Figures and Tables

**Table 1 healthcare-13-00302-t001:** Sociodemographic characteristics of the pregnant women.

	Age (Years)	Education Level	Immigration Time (Years)	GW	History of Miscarriage/Stillbirth	Child Number	Control Visits for Pregnancy
P1	31–35	Middle school	5	22	No	4	Regular
P2	18–25	Primary school	5	40	No	-	Regular
P3	18–25	Primary school	5	34	Miscarriage	2	Regular
P4	18–25	Middle school	10	36	No	**-**	Regular
P5	26–30	University *	5	28	No	2	Regular
P6	18–25	Middle school	9	25	No	1	Regular
P7	18–25	Primary school	3	32	No	2	Regular
P8	31–35	Primary school *	3	24	Miscarriage	4	Regular
P9	31–35	Middle school	10	28	Miscarriage	5	Regular
P10	26–30	University	8	29	No	1	Regular
P11	26–30	Middle school	10	32	No	2	Irregular
P12	18–25	Primary school	10	24	No	1	Regular
P13	26–30	High school *	10	28	No	**-**	Regular
P14	18–25	Middle school *	5	24	Miscarriage and Stillbirth	2	Regular
P15	40–45	Primary school	10	32	Stillbirth	9	Regular
P16	26–30	Primary school	9	39	Stillbirth	3	Regular

P: participant, GW: gestational week, * not graduated.

**Table 2 healthcare-13-00302-t002:** Themes and subthemes.

Themes	Subthemes
Beliefs about vaccines	Perceived benefits about vaccines
Information about vaccines	Need for information
Knowledge, beliefs, and attitudes about tetanus disease and vaccine	Knowledge about tetanus disease
	Knowledge and beliefs about tetanus vaccine
	Attitudes towards tetanus vaccine
Reasons for vaccination desire	Health for pregnant women
	Health for baby/children
	Recommended by health workers
Reasons for not being vaccinated	
Barriers to accessing health services	

## Data Availability

The data presented in this study are available on request from the corresponding author. The data are not publicly available due to privacy and ethical restrictions.

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
