# Peer review of "Tetanus Vaccine Knowledge, Beliefs, and Attitudes Among Syrian Pregnant Women in Türkiye: A Qualitative Study"

_healthcare, 2025, doi:10.3390/healthcare13030302_

Round 1
Reviewer 1 Report
Comments and Suggestions for Authors
To: Healthcare MDPI
Dear EIC,
Dear AE,
This is my review results for the manuscript ID: healthcare-3314897
This is a qualitative study of the general knowledge and beliefs of Syrian migrant or refugee women who live in Turkey. The authors gathered their data through interviews and created some terms and codes for their findings. Finally, a model was used to interpret their findings. Due to the low sample size, I think the results of this study couldn’t add some novel things to the field content.
Comments
· The results section should be shorter. It is full of <2 lined paragraphs. Many of these paragraphs can be inserted in a single table.
· One significant comment is that the study sample size is shallow. The results of this study are not accepted and admitted to a specific part of society.
· Table 2 should be minimized.
· For the content of Table 2, it is better to design a figure to illustrate the terms.
· Kindly remove the section 4.1. and unify the limitations section with the body of the discussion section.
Good luck
Author Response
Point 1: The results section should be shorter. It is full of <2 lined paragraphs. Many of these paragraphs can be inserted in a single table.
Answer 1: Thank you for your valuable suggestion. We have revised the results section by shortening it and merging the two-line paragraphs for better readability and coherence.
Point 2: One significant comment is that the study sample size is shallow. The results of this study are not accepted and admitted to a specific part of society.
Answer 2: As this is a qualitative study, our primary aim was not to achieve statistical representativeness but to gain an in-depth understanding of the topic under investigation. In qualitative research, sample size is determined by data saturation, which occurs when no new themes or insights emerge during the interviews. In our study, we reached data saturation, which guided the completion of our interviews. However, we acknowledge the importance of examining this topic with larger samples in quantitative studies.
Point 3: Table 2 should be minimized. For the content of Table 2, it is better to design a figure to illustrate the terms.
Answer 3: Thank you for your valuable feedback. We have minimized Table 2 as suggested and decided to retain it in a simplified form instead of designing a figure. In this revised version, we removed the last column ("Codes") for brevity and to enhance readability. The updated table now focuses on the main themes and subthemes to provide a clear and concise presentation of the information.
Point 4: Kindly remove the section 4.1. and unify the limitations section with the body of the discussion section.
Answer 4 : Thank you for your insightful feedback. We will remove Section 4.1 as requested and integrate the limitations into the body of the discussion section.
Reviewer 2 Report
Comments and Suggestions for Authors
Dear Authors,
I would like to thank you for your manuscript entitled "Tetanus Vaccine Knowledge, Beliefs, and Attitudes among Syrian Pregnant Women in Türkiye: A Qualitative Study"
I have a major comment:
In order to make a conclusive study, we have to adopt some statistics to make the results clearer and significant in meaning. I can not find any survey statistics here, it is just a story of cases.
Please rewrite your manuscript in this context.
Thank you.
Author Response
Point 1: In order to make a conclusive study, we have to adopt some statistics to make the results clearer and significant in meaning. I can not find any survey statistics here, it is just a story of cases. Please rewrite your manuscript in this context.
Answer 1: Thank you for your valuable feedback and suggestions regarding our manuscript. We would like to clarify that our study was designed as a qualitative research project using semi-structured interview questions to conduct in-depth interviews. As such, no surveys or statistical analyses were employed. Instead, we performed thematic analysis to interpret the data and incorporated participant quotes to support and enrich the findings.
Reviewer 3 Report
Comments and Suggestions for Authors
The study, titled "Tetanus Vaccine Knowledge, Beliefs, and Attitudes among Syrian Pregnant Women in Türkiye: A Qualitative Study," looks into the critical link between healthcare and displacement, shedding light on awareness, perceptions, and barriers to tetanus immunization among pregnant Syrian refugees in Türkiye. This qualitative research offers a thorough understanding of the influence of migration and socio-cultural variables on vaccine uptake, which has substantial implications for public health strategies in refugee settings.
The research aims to identify gaps in pregnant Syrian women's knowledge, beliefs, and attitudes around tetanus immunizations in Turkey. The authors use semi-structured interviews and focus group sessions to capture human stories and cultural nuances expertly. The qualitative technique effectively investigates complex social and psychological issues, such as confidence in healthcare systems and the effects of religious or cultural beliefs.
The volunteers, drawn from refugee clinics, provide diverse perspectives on a group that is sometimes disregarded in health research. The study correctly stresses tetanus, a preventable disease that seriously affects maternal and newborn health. Nonetheless, the authors might have given more information on the specific inclusion criteria and methods used to calculate the sample size for representativeness.
The findings show significant inadequacies in understanding the purpose and timing of tetanus immunization. A large proportion of people were unaware of the link between maternal immunization and neonatal protection, highlighting the urgent need for targeted educational initiatives. Misinformation commonly impacts women's perceptions of the vaccine's safety and efficacy, with some expressing concerns about potential side effects or the vaccine's requirements in light of earlier immunizations.
Vaccination views were influenced by cultural, religious, and environmental factors. Some people considered vaccination as an important preventive measure, while others were hesitant due to distrust of healthcare providers or a lack of communication in their native language. Healthcare accessibility proved to be a significant barrier, compounded by language challenges and financial constraints.
The study's strength is its qualitative approach, which allows for a thorough understanding of the relationship between knowledge, beliefs, and attitudes. The authors provide a detailed theme analysis, carefully categorizing vaccine barriers and facilitators. Their focus on refugee demography is particularly commendable since it highlights the intersection of health disparities and migration.
The study also makes practical suggestions. It highlights the need for culturally appropriate health communication and the necessity for multilingual healthcare personnel. These standards are consistent with worldwide programs focused on improving healthcare justice for displaced groups.
The study makes major contributions; nonetheless, several limitations must be addressed. The restricted sample size and qualitative technique, although useful, limit the application of the findings. Incorporating quantitative features, such as surveys, may improve understanding. The study should look at how overarching systemic variables, such as Turkey's refugee healthcare policy, affect vaccine accessibility.
This study reveals major gaps in tetanus vaccine knowledge and acceptance among pregnant Syrian women in Turkey, providing a foundation for improving immunization rates in similar scenarios. Healthcare practitioners may better support disadvantaged persons by addressing issues such as misinformation, language difficulty, and accessibility constraints. The authors successfully emphasize the need for culturally sensitive and context-specific interventions, so contributing to larger efforts aiming at improving refugee health outcomes.
Author Response
Point 1: The volunteers, drawn from refugee clinics, provide diverse perspectives on a group that is sometimes disregarded in health research. The study correctly stresses tetanus, a preventable disease that seriously affects maternal and newborn health. Nonetheless, the authors might have given more information on the specific inclusion criteria and methods used to calculate the sample size for representativeness.
Answer 1: Dear Reviewer, thank you for your valuable comments. The inclusion criteria for this study were Syrian migrant pregnant women aged 18 years or older who visited the MHC(Migrant health Center) for pregnancy follow-ups or other healthcare services. Participants were selected among those who were informed about the study and provided their consent to participate. This information is added to study. As this is a qualitative study, our primary aim was not to achieve statistical representativeness but to gain an in-depth understanding of the topic under investigation. In qualitative research, sample size is determined by data saturation, which occurs when no new themes or insights emerge during the interviews. In our study, we reached data saturation, which guided the completion of our interviews. However, we acknowledge the importance of examining this topic with larger samples in quantitative studies.
Point 2: The findings show significant inadequacies in understanding the purpose and timing of tetanus immunization. A large proportion of people were unaware of the link between maternal immunization and neonatal protection, highlighting the urgent need for targeted educational initiatives. Misinformation commonly impacts women's perceptions of the vaccine's safety and efficacy, with some expressing concerns about potential side effects or the vaccine's requirements in light of earlier immunizations.
Answer 2: Thank you for your valuable comments.
Point 3: Vaccination views were influenced by cultural, religious, and environmental factors. Some people considered vaccination as an important preventive measure, while others were hesitant due to distrust of healthcare providers or a lack of communication in their native language. Healthcare accessibility proved to be a significant barrier, compounded by language challenges and financial constraints.
Answer 3: Thank you for your valuable comments.
Point 4: The study's strength is its qualitative approach, which allows for a thorough understanding of the relationship between knowledge, beliefs, and attitudes. The authors provide a detailed theme analysis, carefully categorizing vaccine barriers and facilitators. Their focus on refugee demography is particularly commendable since it highlights the intersection of health disparities and migration.
Answer 4: We greatly appreciate your detailed and helpful comments.
Point 5: The study also makes practical suggestions. It highlights the need for culturally appropriate health communication and the necessity for multilingual healthcare personnel. These standards are consistent with worldwide programs focused on improving healthcare justice for displaced groups.
Answer 5: Thank you for your valuable comments.
Point 6: The study makes major contributions; nonetheless, several limitations must be addressed. The restricted sample size and qualitative technique, although useful, limit the application of the findings. Incorporating quantitative features, such as surveys, may improve understanding.
Answer 6: Dear Reviewer, thank you for your valuable comments. As this is a qualitative study, our primary aim was not to achieve statistical representativeness but to gain an in-depth understanding of the topic under investigation. In qualitative research, sample size is determined by data saturation, which occurs when no new themes or insights emerge during the interviews. In our study, we reached data saturation, which guided the completion of our interviews. However, we acknowledge the importance of examining this topic with larger samples in quantitative studies.
Point 7: The study should look at how overarching systemic variables, such as Turkey's refugee healthcare policy, affect vaccine accessibility.
Answer 7: Thank you for your insightful suggestion. As our data collection process has already been completed, this specific aspect could not be addressed in the current study. However, we acknowledge the importance of investigating systemic variables, such as healthcare policies, in influencing vaccine accessibility. We plan to explore these issues further in future studies using quantitative methods
Point 8: This study reveals major gaps in tetanus vaccine knowledge and acceptance among pregnant Syrian women in Turkey, providing a foundation for improving immunization rates in similar scenarios. Healthcare practitioners may better support disadvantaged persons by addressing issues such as misinformation, language difficulty, and accessibility constraints. The authors successfully emphasize the need for culturally sensitive and context-specific interventions, so contributing to larger efforts aiming at improving refugee health outcomes.
Answer 8: Thank you for your encouraging feedback. Your comments further motivate us to continue our efforts in designing interventions aimed at improving refugee health outcomes.
Round 2
Reviewer 1 Report
Comments and Suggestions for Authors
Dear EIC,
Dear AE,
I read the manuscript and was convinced by the corrections.
Author Response
Dear Reviewer,
Thank you for your valuable comment.
Kind regards.
Reviewer 2 Report
Comments and Suggestions for Authors
Dear Authors,
I think you can make ordinal statistics based on Yes or No answers.
Since Tetanus vaccine is well known globally. I was looking for some novelty to gain interest to readers.
Thank you
Author Response
Point 1: Dear Authors,
I think you can make ordinal statistics based on Yes or No answers.
Since Tetanus vaccine is well known globally. I was looking for some novelty to gain interest to readers.
Answer 1:
Dear Reviewer,
Thank you for your insightful comments and suggestions. Our study is a qualitative research that relies on in-depth interviews with 16 participants. The primary aim was to explore participants' perspectives and experiences regarding the Tetanus vaccine rather than to quantify responses. To analyze the data, we employed thematic analysis, which is widely accepted in qualitative studies for its ability to uncover underlying themes and patterns within narratives. Kind regards.